# Roles of Aging, Circular RNAs, and RNA Editing in the Pathogenesis of Amyotrophic Lateral Sclerosis: Potential Biomarkers and Therapeutic Targets

**DOI:** 10.3390/cells12101443

**Published:** 2023-05-22

**Authors:** Takashi Hosaka, Hiroshi Tsuji, Shin Kwak

**Affiliations:** 1Department of Neurology, Division of Clinical Medicine, Faculty of Medicine, University of Tsukuba, Tsukuba 305-8575, Japan; 2University of Tsukuba Hospital/Jichi Medical University Joint Ibaraki Western Regional Clinical Education Center, Chikusei 308-0813, Japan; 3Department of Internal Medicine, Ibaraki Western Medical Center, Chikusei 308-0813, Japan; 4Department of Neurology, Tokyo Medical University, Tokyo 160-0023, Japan

**Keywords:** amyotrophic lateral sclerosis, brain aging, circular RNA, RNA editing

## Abstract

Amyotrophic lateral sclerosis (ALS) is an incurable motor neuron disease caused by upper and lower motor neuron death. Despite advances in our understanding of ALS pathogenesis, effective treatment for this fatal disease remains elusive. As aging is a major risk factor for ALS, age-related molecular changes may provide clues for the development of new therapeutic strategies. Dysregulation of age-dependent RNA metabolism plays a pivotal role in the pathogenesis of ALS. In addition, failure of RNA editing at the glutamine/arginine (Q/R) site of GluA2 mRNA causes excitotoxicity due to excessive Ca^2+^ influx through Ca^2+^-permeable α-amino-3-hydroxy-5-methyl-4-isoxazole propionic acid receptors, which is recognized as an underlying mechanism of motor neuron death in ALS. Circular RNAs (circRNAs), a circular form of cognate RNA generated by back-splicing, are abundant in the brain and accumulate with age. Hence, they are assumed to play a role in neurodegeneration. Emerging evidence has demonstrated that age-related dysregulation of RNA editing and changes in circRNA expression are involved in ALS pathogenesis. Herein, we review the potential associations between age-dependent changes in circRNAs and RNA editing, and discuss the possibility of developing new therapies and biomarkers for ALS based on age-related changes in circRNAs and dysregulation of RNA editing.

## 1. Introduction

Amyotrophic lateral sclerosis (ALS) is characterized by progressive muscle weakness and atrophy due to degeneration of both upper and lower motor neurons, resulting in death from respiratory failure within 2–4 years of diagnosis [1,2]. Since the risk of developing ALS increases drastically with age, the worldwide trend of increased longevity is likely to contribute to the global rise in the incidence of ALS [3]. Although various plausible mechanisms, such as disruption of RNA metabolism, excitotoxicity due to dysregulation of glutamatergic signaling, epigenetic modification, and dysfunction of the endoplasmic reticulum (ER) and mitochondria, have been proposed for the etiology of ALS, the mechanisms underlying motor neuron death in patients with ALS remain elusive [4]. Recent advances in next-generation sequencing have identified at least 40 ALS-linked genes, including transactive response DNA binding protein (*TARDBP*), fused in sarcoma (*FUS*)*,* and chromosome 9 open reading frame 72 (*C9ORF72*) [5]. Most ALS-linked genes encode proteins related to RNA metabolism, suggesting that the disruption of RNA metabolism plays a key role in ALS pathogenesis [6]. Unfortunately, most high-profile clinical trials for ALS have yielded insufficient results [7], and three disease-modifying drugs (riluzole, edaravone, and sodium phenylbutyrate/ursodoxicoltaurine) approved by the United States Food and Drug Administration (FDA) [8,9,10]) do not effectively increase the life expectancy of patients with ALS [2]. Therefore, to develop a definitive treatment for ALS, a novel therapeutic approach based on the pathogenesis of ALS and the establishment of biomarkers for its diagnosis or treatment efficacy are needed.

Aging is an unavoidable process that causes substantial alterations in gene expression in organs and tissues, including the central nervous system (CNS) [11,12,13]. Notably, several markers of aging, including genomic instability, epigenetic alterations, and cellular senescence, have been linked to neurodegeneration [14], suggesting that aging is a primary risk factor for neurodegenerative diseases, including ALS [14]. Therefore, elucidation of the underlying mechanisms of aging and age-associated onset and progression of ALS would help clarify the pathogenesis of ALS.

Circular RNAs (circRNAs) are single-stranded RNA molecules circularized by the covalent joining of the 3′-end to the 5′-end, known as back-splicing [15,16] and can be divided into three classes: circular intronic RNA (ciRNA), exonic circRNA (ecircRNA), and exon–intron circRNA (EIciRNA) [17]. CiRNAs are produced by canonical splicing and escape from debranching enzyme, whereas ecircRNA and EIciRNAs are generated by back-splicing with the assistance of complementary sequences between flanking intron and RNA-binding proteins (RBPs) [18]. As their characteristic structure renders them resistant to degradation from the RNA decay machinery, circRNAs are highly stable compared to their cognate RNAs. Notably, expression levels of circRNAs exhibit tissue-, development-, and sex-specific patterns in mammals independent of their cognate RNAs [19,20,21,22,23,24]. CircRNAs play important roles in modulating a variety of biological processes in the nucleus and cytoplasm, and are involved in the regulation of transcription, alternative splicing of pre-mRNA, and chromatin looping in the nucleus [25,26], while acting as sponges of microRNAs (miRNAs) and RBPs in the cytoplasm, thereby regulating translation through the prevention of binding between miRNAs or RBPs and their target RNAs [22,27,28]. CircRNAs are highly enriched in the brain, and their levels are drastically altered in advanced age and neurodegenerative diseases [23,29,30]. Additionally, some circRNAs are involved in the processing of several age-associated markers, including cellular senescence and epigenetic modification [31,32]. These findings suggest that circRNAs are involved in the pathogenesis of neurodegenerative diseases, for which brain aging is a risk factor, and are targets for therapy [14,33]. As circRNAs cross the blood–brain barrier and are found in body fluids, such as the cerebrospinal fluid (CSF), serum, and plasma [34,35], their presence in body fluids can be potential biomarkers for neurodegenerative diseases [31].

Adenosine-to-inosine conversion of RNA (A-to-I RNA editing), a post- or cotranscriptional modification of RNA catalyzed by adenosine deaminase acting on RNA (ADARs), occurs in various classes of RNAs, including miRNAs and circRNAs, and plays an important role in complex CNS functions [36]. RNA editing of intronic regions affects RNA splicing and the biogenesis of circRNAs, whereas editing of exonic regions or miRNAs affects the translation, localization, and stability of RNA [37,38,39,40]. Excitotoxicity resulting from the dysregulation of glutamatergic signaling via excessive Ca^2+^ influx through the alpha-amino-3-hydroxy-5-methyl-4-isoxazole propionic acid (AMPA) receptor is a plausible mechanism underlying motor neuron death in patients with ALS [41]. The Ca^2+^ permeability of AMPA receptors depends on the presence or absence of a glutamine/arginine (Q/R) site-edited GluA2 subunit [42], indicating that the dysregulation of RNA editing is involved in the pathogenesis of ALS. Aging influences expression levels of ADARs and the editing efficiency at some editing sites [43,44]. *ADARB1* and *ADARB2*, which encode ADAR2 and ADAR3, respectively, are longevity genes [45]. Moreover, the expression of postsynaptic AMPA receptors is reduced with aging due to the increased elimination of hypofunctional AMPA receptors resulting from the age-dependent reduction of positive allosteric modulators [46,47].

Age-dependent dysregulation of RNA editing may contribute to the age-dependent accumulation of circRNAs in the brain [29,43], which may play a pathogenic role in ALS. Although there have been many comprehensive reviews conducted that have demonstrated circRNAs or RNA modification to be associated with age-related neurodegenerative diseases including ALS [48], no reviews have described the role of aging, circular RNAs, and RNA editing in the pathogenesis of ALS. To the best of our knowledge, this is the first review which describes age-related alterations in circRNAs and RNA editing, focusing on their potential roles in neurodegeneration and the pathogenesis of ALS, and on the possibility of circRNAs and the dysregulation of RNA editing as biomarker candidates and therapeutic targets for ALS.

## 2. Age-Related Changes of circRNAs and RNA Editing

### 2.1. Age-Related circRNAs in the Brain (Figure 1)

Cellular senescence, a stress-induced state of indefinite growth arrest that increases with age, plays a role in maintaining the survival of healthy cells, facilitates the removal of damaged cells [13,33], and is an antagonistic response to the primary damage of cells and a marker of brain aging [14]. Forkhead box O3 (*FOXO3*), a transcription factor that plays a critical role in brain development and aging, is a longevity gene and is implicated as a causative gene of neurodegenerative diseases [49]. CircFOXO3, the circular form of *FOXO3*, sequesters antistress or antisenescence proteins, thereby arresting the cell cycle and cell proliferation in concert with p21 and cyclin-dependent kinase 2 [28,50]. Increased expression of circFOXO3 is involved in cellular senescence in the CNS, which may lead to neurodegeneration; however, a controversial report has shown that expression levels of circFOXO3 are significantly decreased in the blood of elderly persons and in late passage primary culture cells [51]. CircPVT1, a circular form of an exon of *PVT1*, influences cellular senescence and neurodegeneration by changing the expression levels of let-7 and miR-199-5a: let-7 regulates immune response, autophagy, and apoptosis [52,53,54], whereas miR199-5a regulates the expression levels of sirtuin1 (SIRT1) mRNA, which is associated with brain aging and neurodegeneration resulting from the dysregulation of mitochondrial energy metabolism [55,56,57,58].

**Figure 1 cells-12-01443-f001:**
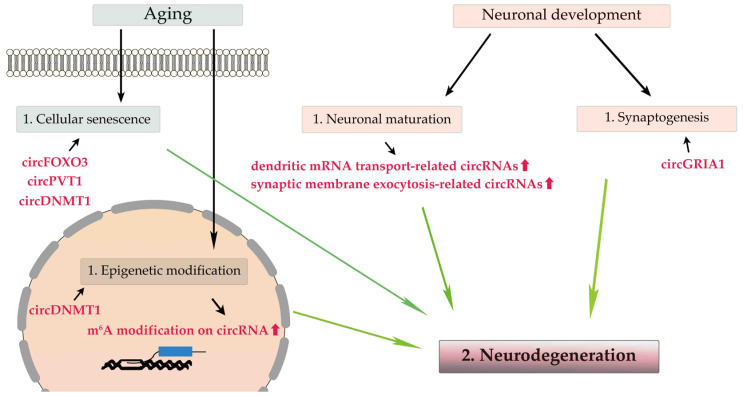
Age-related changes in circRNAs within the brain. Aging causes cellular senescence and epigenetic modification, and neuronal development involves neuronal maturation and synaptogenesis. CircRNAs, including circFOXO3, circPVT1, and circDNMT1, are involved in cellular senescence, and epigenetic m^6^A modification of circRNAs increases with aging. During neuronal maturation, the expression levels of circRNAs associated with dendric mRNA transport, synaptic membrane exocytosis, and synaptogenesis is increased.

Epigenetic modifications, such as DNA and RNA methylation, influence chromatin function, and their age-related changes are associated with brain aging and neurodegeneration [14]. RNA methylation is a major form of epigenetic RNA modification, and the most common site is N^6^-methyladenosine (m^6^A) [59,60]. CircRNAs regulate m^6^A modification, and conversely, an age-dependent increase in m^6^A modification affects the biogenesis, stability, translation, and biological function of circRNAs [60]. Significant differences in the m^6^A modification of several circRNAs in Alzheimer’s disease model mice compared with control mice [61] suggest the possible involvement of the m^6^A modification of circRNAs in neurodegeneration, which still needs to be convincingly demonstrated. DNA methyl-transferase 1 (DNMT1)-mediated DNA hypermethylation regulates age-dependent cell death in the brain [62]. Accumulating evidence has indicated that DNMT1 plays a crucial role in the survival and maturation of various classes of neurons [63,64,65], and circDNMT1 (has_circ_102439), a circular form of *DNMT1*, influences DNMT1 mRNA stability and cellular senescence by affecting autophagy [66].

During neuronal maturation, the expression levels of most circRNAs, especially cognate mRNAs that are translated into proteins related to dendritic mRNA transport and synaptic membrane exocytosis, are upregulated without correlation to their cognate RNAs’ expression levels [23,31]. Additionally, during synaptogenesis, many circRNAs expressed in the brain are enriched in the synaptic space, and their expression levels are altered regardless of their cognate linear mRNAs’ expression levels [23,30]. Therefore, circRNAs play a pivotal role in neuronal maturation and synaptogenesis independent of their cognate mRNAs [20,23], and perturbation of circRNA expression may be a cause of neurodegeneration. Indeed, circGRIA1, a circular isoform of the AMPA receptor subunit *GRIA1*, exhibits age-associated and male-specific upregulation in the prefrontal cortex and hippocampus of rhesus macaques, and circGRIA1 knockdown leads to an improvement in age-related synaptic plasticity of hippocampal neurons, suggesting a role for circGRIA1 in age-associated synaptic decline [67].

Although comprehensive analyses of human and rhesus macaque samples have demonstrated age-dependent changes in the expression levels of several circRNAs, the biological significance of these alterations in brain aging and neurodegeneration remains unclear [51,68,69].

### 2.2. Age-Related RNA Editing

Among the three ADAR proteins expressed in mammals, ADAR1 and ADAR2 have essential editing activities. ADAR1 is ubiquitously expressed and is primarily responsible for RNA editing in repeat elements in noncoding regions of mRNAs, whereas ADAR2, which is expressed highest in the CNS, is involved in recoding and editing in protein-coding regions [36,70]. ADAR3, which is highly enriched in oligodendrocytes in the brain, lacks editing activity and instead acts as a negative regulator of RNA editing by sequestering the editing substrates of ADAR1 and ADAR2 [36,71,72]. RNA editing within protein-coding sequences alters protein structure and function, leading to alterations in multiple biological processes, including synaptic transmission and immune responses [37,73,74]. RNA editing is increased during development in the mammalian brain [75,76] and, conversely, decreased in age-related diseases [77,78]. These findings suggest a modulatory role for RNA editing in human aging.

Single-nucleotide polymorphisms (SNPs) in *ADARB1* and *ADARB2* are associated with exceptional longevity. The frequencies of common alleles for 18 SNPs in *ADARB1* and *ADARB2* are increased in the oldest patients and, among them, those for SNP rs3788157 in *ADARB1* and rs17294019 in *ADARB2* tend to increase with age [45]. In addition, the inactivation of *adr-1* and *adr-2*, encoded by *Caenorhabditis elegans’* respective orthologs of *ADARB1* and *ADARB2*, reduces lifespan [45], whereas mutations in *ADAR* or *ADARB1* are associated with neurodevelopmental disorders, such as developmental epileptic encephalopathy and bilateral striatal necrosis [79,80,81]. These results suggest that ADARs play a role in both brain development and aging.

The editing efficiency at some sites decreases with age. The editing efficiency at the lysine/glutamic acid (K/E) site of cytoplasmic fragile X mental retardation protein-interacting protein 2 (*CYFIP2*) mRNA is decreased in an age-dependent manner in the human cortex [82,83]. Moreover, the editing efficiency at the Q/R site of GluA2 mRNA is reduced in an age-dependent manner in mouse motor neurons [43]. Additionally, expression levels of RNA-editing enzymes decrease with aging. Expression levels of ADAR1 mRNA are reduced in the brain cortex, and expression levels of ADAR2 mRNA in the anterior horns of the spinal cord, but not in the brain cortex of aged mice, are reduced [43,44].

## 3. ALS-Related Changes of circRNAs and Dysregulation of RNA Editing

### 3.1. ALS-Related circRNA

CircRNAs are associated with brain aging and neurodegeneration, and the presence of disease-specific circRNAs has been reported in the brains of patients with Alzheimer’s or Parkinson’s disease [84] as well as in the spinal cord and muscles of patients with ALS [85,86]. As no studies have demonstrated the direct involvement of circRNAs in ALS pathogenesis, we discuss evidence suggesting a role for circRNAs in the pathogenesis of ALS (Figure 2).

Several ALS-linked genes encode RBPs, and the aberrant protein aggregation of RBPs is a pathological characteristic of motor neurons in ALS. Therefore, the resulting disruption of RNA metabolism plays a key role in ALS pathogenesis [6]. FUS plays a critical role in splicing regulation, and subcellular mislocalization of FUS leads to cell death-causing aberrant RNA metabolism in ALS motor neurons [87]. The biogenesis of circRNAs is affected by the interaction of FUS with intron-flanking back-splicing junctions without significant effects on the expression of cognate linear RNAs [88]. Additionally, the expression levels of several circRNAs are altered in induced pluripotent stem cell (iPSC)-derived motor neurons of patients with ALS carrying the *FUS*^P525L^ mutation compared with those carrying *FUS*^WT^ [89], although the pathogenic significance and effects on motor neuron biology of the circRNA expression changes in ALS patients carrying the *FUS* mutation remain unknown. The intronic hexanucleotide (GGGGCC) repeat expansion (HRE) of *C9orf72* is the most common genetic cause of ALS in Europe and America [90,91] in which pathogenic roles of non-AUG translation-mediated production of toxic dipeptide repeat (DPR) proteins and sequestration of RBP in nuclear RNA granules have been hypothesized [92]. Notably, intron-derived circRNA, including the HRE of *C9orf72,* is translated into toxic DPR proteins [93].

Evidence suggests a role for circRNAs in the epigenetic modification of nucleic acids in ALS [94]. Methyl-CpG binding domain protein 2 (MDB2) binds to a fraction of hypomethylated genes and plays a pivotal role in methylation-related transcription regulation [95]. Knockdown of circKCNN2 (has_circ_0127664), a circular form of potassium calcium-activated channel subfamily N member 2 (*KCNN2*), leads to the downregulation of MDB2 [96]. Expression levels of circKCNN2 are considerably reduced in the cortical neurons of patients with frontotemporal dementia, exhibiting mislocalization of the transactive response DNA-binding protein of 43 kDa (TDP-43) from the nucleus to the cytoplasm (TDP-43 pathology) as compared with those in control subjects [97]. Moreover, some circRNAs influence the expression levels of epigenetic enzymes; circELP3 (hsa_circ_0001785) regulates the expression of elongator acetyltransferase complex subunit 3 (ELP3) [98], a histone acetyltransferase, that has an association with motor neuron degeneration via regulating heat shock protein [99]. CircTHBS2 (hsa_circ_0078710) and circSIRT1 (hsa_circ_0093844) regulate the expression levels of histone deacetylases (HDACs) and SIRT1, respectively [100,101], both of which have an association with epigenetic modification in ALS pathogenesis [102,103]. CircLRP6 regulates the expression level of protein arginine N-methyltransferase (PRMT1), a histone methyltransferase [104], that rescues neurite growth in *FUS^R521C^* mice [105]. However, there is no evidence supporting the direct roles of these circRNA-related expression changes in neurodegeneration or ALS pathogenesis.

A pathogenic role for excitotoxicity resulting from excessive Ca^2+^ influx into motor neurons by GluA2-lacking Ca^2+^-permeable AMPA receptor ion channels has been proposed in ALS [106,107]. AMPA receptors are homo- or hetero-tetramers of GluA1–GluA4 subunits. Their tightly regulated biogenesis, membrane trafficking, and degradation result in well-regulated physiological CNS activity [108]. The upregulation of GluA1 mRNA, which reduces the proportion of the GluA2 subunit among the four subunits, is associated with excitotoxicity in the spinal cord and iPSC-derived motor neurons of patients with *C9orf72* ALS [109] and *FUS* knockdown mice [110,111,112,113]. CircGRIA1 negatively regulates the expression levels of GluA1 mRNA and protein expression by competitively binding to the promotor region of *GRIA1* [67].

The dysfunction of the ER and mitochondria due to the alteration of ER–mitochondrial signaling is another hypothesis for ALS pathogenesis [114]. The ER physically contacts the mitochondria through specialized lesions called mitochondria-associated membranes (MAMs), and studies have reported an association between MAM disruption and the pathogenesis of various neurodegenerative diseases, including ALS [115,116,117]. Mutations in sigma nonopioid intracellular receptor 1 (*SIGMAR1*), which encodes the sigma-1 receptor (Sig1R), cause juvenile ALS (ALS16), and mutant Sig1R loses its MAM-specific chaperone protein function [118]. Overexpression of *Sig1R^E102Q^* mutant proteins induces neuronal cell death [119], and loss of wild-type Sig1R proteins induces the collapse of MAMs in the motor neurons of *Sig1R^−/−^* mice [115]. The significance of changes in the expression levels of SigR1 in ALS has been inconsistently reported; expression levels of mutant Sig1R proteins (c672*51G > T) are either elevated in leukocytes and the frontal cortex [120] or not different in primary lymphoblastoid cells derived from patients with ALS carrying mutant *Sig1R^E102Q^* [121]. In addition, circHIPK2, a circular form of homeodomain-interacting protein kinase 2 (*HIPK2*), affects the expression levels of Sig1R by acting as a sponge for miR124-2HG [122].

Taken together, although changes in the expression of circRNAs have been proposed to be associated with ALS pathogenesis-related molecular change, direct evidence suggesting the role of the expression alteration of circRNAs in ALS pathogenesis remains elusive. Moreover, although rno_circ_013017 inhibits motor neuron apoptosis in rats with spinal cord injury [123], the role of circRNA in motor neuron biology in healthy elderly people or ALS patients has not been reported. Further knowledge on the etiological roles of circRNAs in motor neuron death in ALS will elucidate the interplay of several circRNAs in brain aging and neurodegeneration.

### 3.2. Dysregulation of RNA Editing in ALS Motor Neurons (Table 1)

Evidence that elevated glutamate levels in the postmortem tissue and CSF of patients with ALS [124,125,126,127], the loss of high-affinity glutamate uptake [128], and riluzole, an inhibitor of glutamate release, improve one-year survival rates, especially in the late stages of ALS [8,129,130,131] has implicated excitotoxicity as a cause of ALS pathogenesis. Among the subtypes of glutamate receptors, Ca^2+^-permeable AMPA receptors specifically mediate the slow death of motor neurons, and the increase in their Ca^2+^ permeability results from the incorporation of the Q/R site-unedited GluA2 subunit into their assembly. Adenosine at the Q/R site of GluA2 pre-mRNA is specifically converted to inosine by ADAR2, and the edited Q/R site of GluA2 mRNA is translated into arginine (R; codon CGG) but not into glutamine (CAG; the genomic codon) because inosine in mRNA is recognized as guanosine during translation. As mammalian CNS neurons express only Q/R site-edited GluA2 and the majority of AMPA receptors expressed in the synapses of CNS neurons contain GluA2 in their assembly, the synaptic AMPA receptors are not Ca^2+^-permeable [132,133,134].

In the spinal motor neurons of patients with sporadic ALS, Q/R site-unedited GluA2 is expressed because of the downregulation of ADAR2 [135,136]. Motor neuron-specific conditional ADAR2 knockout mice (ADAR2*^flox^*^/*flox*^/VAChT. Cre; AR2 mice) exhibit progressive motor dysfunction with degeneration of motor neurons, resulting from excessive Ca^2+^ influx into motor neurons through Ca^2+^-permeable AMPA receptors that have Q/R site-unedited GluA2 subunits [137,138,139]. The death cascade initiated by ADAR2 downregulation is specific to the motor neurons of patients with ALS and is not observed in other neurons of patients with ALS or in the motor neurons of normal control subjects or patients with other neurological diseases [135,136,140]. Moreover, TDP-43 pathology is exclusively observed in the spinal motor neurons that lack ADAR2 immunoreactivity in patients with ALS [129]. ADAR2-lacking motor neurons in AR2 mice exhibit TDP-43 pathology-like mislocalization of TDP-43 resulting from the cleavage of TDP-43 into aggregation-prone fragments by continuous activation of calpain in the cytoplasm due to excessive Ca^2+^ influx [138]. This is likely the mechanism underlying the formation of TDP-43 pathology exclusively in ADAR2-lacking motor neurons in patients with sporadic ALS [139,141]. Therefore, the dysregulation of ADAR2 in motor neurons is likely a disease-causing and disease-specific molecular abnormality in sporadic ALS. Recently, sodium phenylbutyrate/ursodoxicoltaurine, a combination drug of HDAC inhibitor and activator of mitochondria bioenergetic, was approved as an ALS treatment by Health Canada and the FDA in United States [10]. This suggests that epigenetic modification such as histone modification and chromatin remodeling enzyme may contribute to ALS pathogenesis [142]. Intriguingly, inhibition of HDACs by treatment with trichostatin A increases the expression level of ADAR2 mRNA [143].

Furthermore, reduced ADAR2 activity is found in some familial ALS cases as well as in most sporadic ALS cases. The presence of Q/R site-unedited GluA2 due to downregulation of ADAR2 has been observed in motor neurons of patients with ALS carrying the *FUS*^P525L^ mutation [144], and reduced ADAR2 activity due to ADAR2 mislocalization or binding with poly-PR and widespread reduction in RNA editing has been found in the motor neurons and iPSC-derived motor neurons of patients with ALS carrying *C9orf72* with enhanced HRE [145,146].

Collectively, as RNA editing plays a pivotal role in brain aging and in the pathogenesis of sporadic ALS, as well as some forms of familial ALS, dysregulation of RNA editing might provide clues for the identification of biomarkers and serve as a potential therapeutic target for both brain aging and ALS (Table 1).

**Table 1 cells-12-01443-t001:** Etiological role of the dysregulation of RNA editing.

ALS Type	Dysregulation of RNA Editing	Relation to Disease Pathogenesis	Pathogenetic Alteration	Influence on circRNA	Reference
Sporadic	ADAR2 downregulation Reduction of editing efficiency at the Q/R site in GluA2	Excitotoxicity due to exaggerated Ca^2+^ influx	Neuronal death TDP-43 mislocalization	Not described	[135,138]
*FUS*^P525L^ mutation	ADAR2 downregulation	Not described	FUS mislocalization	Alteration of the expression level of several circRNAs	[89,144]
*C9ORF72* with enhanced HRE	Reduction of ADAR2 activity	Not described	ADAR2 mislocalization Poly PR binds to ADAR2	Intron-derived circRNA is translated into toxic DPR proteins	[93,145,146]

Abbreviations: ALS—amyotrophic lateral sclerosis; ADAR2—adenosine deaminase acting on RNA2; TDP-43—transactive response DNA-binding protein of 43 kDa; FUS—fused in sarcoma; HRE—hexanucleotide repeat expansion; DPR—dipeptide repeat.

### 3.3. Aging, circRNAs, and RNA Editing in ALS

Aging is a major risk factor for neurodegenerative diseases, and dysregulation of circRNAs and RNA editing with aging are associated with neurodegenerative diseases, including ALS. Although evidence has demonstrated age-associated changes in RNA editing activity and circRNA processing as described above, only some evidence has demonstrated RNA editing of the exonic region of circRNAs or the role of RNA editing in the biogenesis of circRNAs.

A-to-I RNA editing influences the biogenesis of circRNAs, and the complementary sequence across flanking introns, which contains many Alu repeats, facilitates the formation of circRNAs. Editing sites in Alu repeats are the main targets of ADARs [147,148,149]. The expression levels of circRNAs correlate negatively with those of ADAR1 during neuronal differentiation without modulation of cognate RNAs [23,40,150,151,152]. Similarly, the downregulation of ADAR2 increases the formation of circRNAs in heart tissue and extracellular circRNA levels in the cultured medium of SH-SY5Y cells [153,154].

Although a large fraction of brain circRNA is derived from the exonic coding region [23], whether the A-to-I sites in circRNAs are edited by ADARs, similarly to those in their cognate RNAs, is unclear. We confirmed that the editing efficiency at the Q/R site of circGRIA2 (has_circ_0125620), a circular form of *GRIA2*, changed in parallel with that of the cognate GluA2 mRNA in cultured cells [153]. Therefore, RNA editing of the exonic region of circRNAs may be reduced as RNA editing of their cognate mRNA is dysregulated in motor neurons in sporadic ALS.

Although whether age-related changes in circRNAs influence RNA editing activity and the pathogenesis of ALS is unclear, a close association exists between aging, circRNAs, and the dysregulation of RNA editing (Figure 3). The elucidation of their roles in the pathogenesis of ALS provides crucial insights into novel therapeutic strategies based on the underlying disease-specific molecular abnormalities in ALS.

## 4. CircRNAs and the Dysregulation of RNA Editing as Potential Biomarkers and Therapeutic Targets in ALS

### 4.1. Potential Biomarker Candidates for ALS

CircRNAs are most abundant in the brain and are stable after secretion into body fluids because of their distinctive structure [24,31,155]. As dysregulation of RNA editing increases the formation of circRNAs and extracellular total circRNA levels [153,154], changes in the expression levels of circRNAs could be biomarker candidates for ALS. Several studies have reported comprehensive changes in circRNA expression levels in tissues and sera derived from patients with ALS [85,86,156]. A study has reported reduced expression levels of circPICALM (hsa_circ_0023919) and increased expression levels of circSETD3 (hsa_circ_0000567), circFAM120A (hsa_circ_0005218), circHERC1 (hsa_circ_0035796), circTAF15 (hsa_circ_0043138), circ TNRC6B (hsa_circ_0063411), and circSUSD1 (hsa_circ_0088036) in leukocytes from patients with ALS as compared with healthy control subjects [156]. Among these, hsa_circ_0000567 and hsa_circ_0063411 contain binding sites for miR-9 and miR-641, respectively, the expression levels of which are shown to change in ALS patients [157,158], and hsa_circ_0023919, hsa_circ_0063411, and hsa_circ_0088036 are potential diagnostic biomarkers because of their high sensitivity and specificity to ALS [156,159]. Furthermore, comprehensive studies have reported changes in the expression levels of tens of circRNAs in the spinal cord, cortex, and skeletal muscles of patients with ALS [85,86] and have claimed that these circRNAs could be potential biomarker candidates if they can be detected in body fluids. However, the association between these candidate circRNAs and ALS pathogenesis remains elusive.

As the editing efficiency at the ADAR2-dependent sites in extracellular RNAs correlates with intracellular ADAR2 activity in vitro, changes in ADAR2-dependent editing sites of extracellular RNAs in body fluids, such as CSF, are promising diagnostic biomarkers of ALS. Indeed, Q/R site-unedited GluA2 mRNA and/or circGRIA2 are potential diagnostic biomarkers of ALS [153].

### 4.2. Therapeutic Targets for ALS

A potential role for circRNAs in neurodegeneration has been proposed [32], and a circRNA-based therapeutic strategy, such as the delivery or knockdown of circRNAs, has been put forward for ALS. Based on the hypothesis that accumulated TDP-43 is toxic to neurons, intron-derived circRNAs resulting from the inhibition of the intron debranching enzyme (DBR1), which catalyzes the debranching of lariat introns, were tested, and inhibition of DBR1 was found to suppress the toxicity of TDP-43 in yeast [160]. Moreover, DNA methyl-transferases (DNMTs) inhibitor improves motor function and extends the lifespan of superoxide dismutase 1 (*SOD-1*) mutant ALS model mice [161], in which expression levels of DNMT1 are increased in the spinal cord. Recently, improved expression levels of circRNAs via the extracellular vesicle-mediated delivery of circRNAs were demonstrated [162,163]; therefore, the delivery of circRNAs or siRNA-mediated knockdown to improve the expression levels of circRNAs has been developed as a potential future therapeutic strategy.

The dysregulation of RNA editing due to ADAR2 downregulation can also be a potential therapeutic target for ALS. As the downregulation of ADAR2 explains many aspects of disease-specific pathological changes in sporadic ALS, the restoration of ADAR2 activity and the reduction of excessive Ca^2+^ influx through abnormal AMPA are promising therapeutic strategies for ALS. The delivery of ADAR2 cDNA using a neuron-specific promoter to motor neurons with adeno-associated virus serotype 9 (AAV-9) in AR2 mice, a mouse model of sporadic ALS, markedly suppressed progressive motor dysfunction without adverse effects [164]. Moreover, the injection of the AAV-9 vectors completely prevented progressive motor neuron death and improved TDP-43 mislocalization [164]. As ADAR2 overexpression has no adverse effects on motor, lung, or heart functions, except for simple obesity due to chronic hyperphagia in *ADARB1* transgenic mice [165], ADAR2 activity is safely restored by gene therapy. Although the most recent clinical trials of gene therapy are for familial ALS [166,167,168], a clinical trial of the AAV-ADAR2 vector for sporadic ALS has been initiated (https://www.jichi.ac.jp/hospital/top/consultation/index.html (21 May 2023)). As gene therapies using AAV vectors have been successfully introduced into clinical medicine [169,170,171,172], AAV9-ADAR2 therapy seems to be a promising fundamental treatment for sporadic ALS. When this type of therapy is realized, the need for biomarkers of intracellular ADAR2 activity will be immensely increased.

AMPA receptor antagonists, such as 1,2,3,4-tetrahydro-6-nitro-2,3-dioxo-benzo[f]quinoxaline-7-sulfonamide disodium salt (NBQX) and perampanel, and AMPA receptor-specific RNA aptamers have been shown to protect against the motor neuron death of ALS model animals [173,174,175,176,177]. The administration of NBQX prolongs the survival of transgenic *SOD1^G93A^* mutant mice [174], and the administration of perampanel prevents the progression of the ALS-like phenotype in AR2 mice and increases the cortical excitability threshold in patients with ALS [175,176]. However, in a recent phase 2 clinical trial of perampanel, the inhibition of disease progression was not observed, although there was an improvement in the manual muscle testing score of the lower limbs [178]. Since non-AMPA antagonistic function, such as modulation of voltage-gated sodium channel and regulation of several kinases, with perampanel [179,180] and a dose-dependent increase in drop-out cases due to serious adverse effects might have influenced the results of the phase 2 trial, more selective AMPA receptor antagonists with fewer side effects are required. RNA aptamers targeting AMPA receptor subunits block the exaggerated Ca^2+^ influx, and their administration prevents the progression of motor dysfunction, improves TDP-43 mislocalization, and prevents motor neuron death in AR2 mice [177]. Notably, the lack of sedative effects on the CNS of these AMPA receptor-targeted RNA aptamers makes them potential ALS drugs, as current AMPA receptor inhibitors suppress the activity of all CNS neurons, and suppression of physical brain function is an unavoidable adverse effect of AMPA receptor antagonists in clinical use [177]. Other potential ALS drugs targeting neuronal hyperactivity or excitotoxicity have been reported; memantine, a noncompetitive antagonist of N-methyl-D-aspartic acid (NMDA) receptors, has been shown to delay disease progression and prolong survival of *SOD1* mutant ALS model mice [181] but has not demonstrated therapeutic benefits for ALS patients [182]. Mexiletine, a sodium channel blocker, has been shown to inhibit neuronal hyperexcitability and reduce the frequency of muscle clump [183,184], but it is ineffective in inhibiting disease progression [185]. Ezogabine, an activator of Kv7 potassium channels, has been shown to reduce neuronal excitability in vitro and in vivo [186,187]. There is no report, however, indicating the association between these candidate drugs and the dysregulation of RNA editing or circRNAs.

## 5. Conclusions

Aging, circRNAs, and dysregulation of RNA editing are closely associated with one another. As aging is a major risk factor for neurodegenerative diseases, including ALS, scrutiny of the critical roles of age-dependent circRNAs and dysregulation of RNA editing in the pathogenesis of ALS is required to establish new disease-specific therapeutic approaches and biomarkers for ALS as well as for brain aging.

## Figures and Tables

**Figure 2 cells-12-01443-f002:**
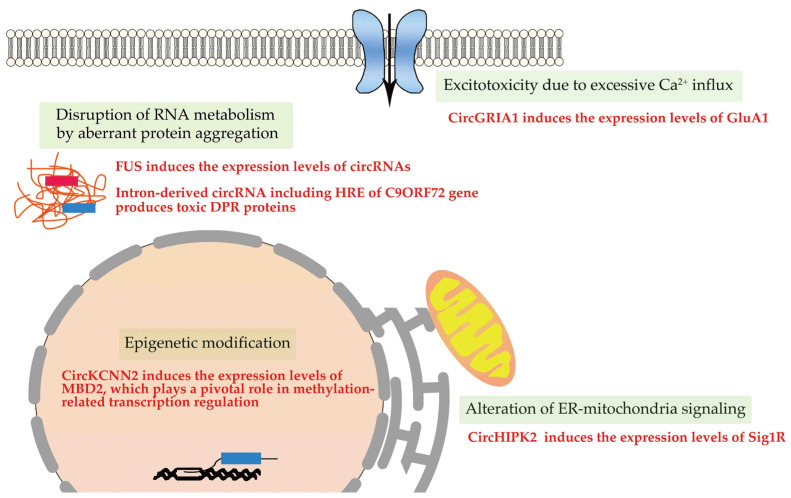
ALS-related changes in circRNAs. Various plausible mechanisms have been proposed for the etiology of ALS. Among them, the role of circRNAs has been reported in the disruption of RNA metabolism by aberrant protein aggregation, epigenetic modification, excitotoxicity due to excessive Ca^2+^ influx, and the alteration of ER–mitochondria signaling.

**Figure 3 cells-12-01443-f003:**
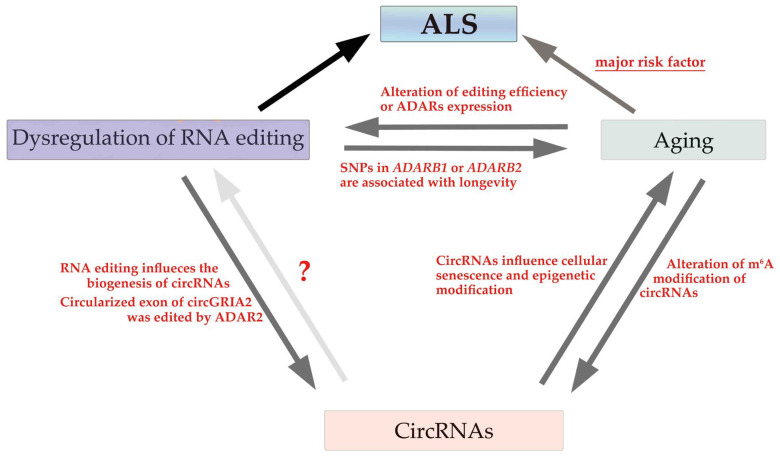
Role of aging, circRNAs, and the dysregulation of RNA editing in ALS pathogenesis. The association between aging and circRNAs or dysregulation of RNA editing has been identified. However, no evidence regarding whether circRNAs influence the dysregulation of RNA editing exists. Moreover, although the involvement of circRNAs in the pathogenesis of ALS has not been demonstrated, aging is a major risk factor for ALS, and the dysregulation of RNA editing is involved in the pathogenesis of ALS.

## Data Availability

Not applicable.

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
