# Peer review of "Roles of Aging, Circular RNAs, and RNA Editing in the Pathogenesis of Amyotrophic Lateral Sclerosis: Potential Biomarkers and Therapeutic Targets"

_cells, 2023, doi:10.3390/cells12101443_

Round 1

Reviewer 1 Report

Hosaka, Tsuji, and Kwak have contributed a review paper on RNA editing and circular RNAs and aging in the pathogenesis of ALS. The topic is important and contemporary. This manuscript is an enjoyable and informative read. The Kwak lab has achieved legendary status regarding RNA editing in ALS. However, though an exciting field, this review has some shortcomings that they might want to consider.

11.       The overall structure of the review seems disorganized. After the introduction, the sections review changes in the expression of circular RNA associated with brain aging and ALS pathogenesis, then RNA editing dysregulation in aging and ALS, then again aging, circular RNAs, and RNA editing. It might be better to have after the Introduction a section on only aging and, then ALS and circular RNAs and RNA editing, and lastly circular RNAs and RNA editing as ALS biomarkers and therapeutic targets.

22.  In the 3 figures, perhaps consider identifying different components with sequential numbers to suggest a stepwise process. Otherwise, the flow of the process is not clear.

33.  The disease causality (pathogenesis) with circular RNAs and RNA editing perturbations are not clearly presented. Please consider designing 2 tables (one for circular RNAs and one for RNA editing) to identify key observations and references that the authors believe to indicate etiological roles in ALS pathogenesis, rather than downstream epiphenomena or secondary to a degenerative process.

44.  Please keep the focus of the review on motor neuron biology as much as possible. Sometimes in the text (for example lines 145-149) it is not at all clear how this information relates to ALS pathogenesis.

55.       The discussion of the anti-excitotoxic therapy testing in ALS is surprisingly not well developed or missing key information. The authors should discuss the trials with memantine, mexiletine, and retigabine.

66.     Because of their work on this, the authors should better note that there were important issues with perampanel trial in that it was not well tolerated by the patients with potential detrimental effect.

77.   The text in general needs some careful editing because of awkward structure.   For example, lines 98-99, 117-122,     

The text in general needs some careful editing because of awkward structure.   For example, lines 98-99, 117-122,     

Author Response

Point-by-Point Responses to Reviewers’ comments

○Reviewer 1

  1. The overall structure of the review seems disorganized. After the introduction, the sections review changes in the expression of circular RNA associated with brain aging and ALS pathogenesis, then RNA editing dysregulation in aging and ALS, then again aging, circular RNAs, and RNA editing. It might be better to have after the Introduction a section on only aging and, then ALS and circular RNAs and RNA editing, and lastly circular RNAs and RNA editing as ALS biomarkers and therapeutic targets.

Response: We really appreciate the reviewer’s constructive suggestions. We have changed sections as follow; 1. Introduction, 2. Age-related changes of circRNAs and RNA editing, 3. ALS-related changes of circRNAs and dysregulation of RNA editing, 4. CircRNAs and dysregulation of RNA editing as potential biomarkers and therapeutic targets in ALS, and 5. Conclusions. Additionally, we have modified our manuscript to make the flow of the text more logical.

  1. In the 3 figures, perhaps consider identifying different components with sequential numbers to suggest a stepwise process. Otherwise, the flow of the process is not clear.

Response: Thank you for this valuable comment. For better understanding of the readers about how aging- and development-related changes influence neuronal degeneration in general, we have modified figure 1. As we focused our discussion on the role of circRNA changes for RNA editing dysregulation, we summarized the so-far known evidence of circRNA changes associated with ALS in figure 2 and their possible interaction with dysregulation of RNA editing, which has a close relevance to ALS pathogenesis, and aging, the major risk factor of ALS, in figure 3. Since evidence suggest that dysregulation of RNA editing due to ADAR2 downregulation influences only circRNAs among other aging-related molecules, we have summarized in the three figures how aging-associated circRNA changes interact with dysregulation of RNA editing.

  1. The disease causality (pathogenesis) with circular RNAs and RNA editing perturbations are not clearly presented. Please consider designing 2 tables (one for circular RNAs and one for RNA editing) to identify key observations and references that the authors believe to indicate etiological roles in ALS pathogenesis, rather than downstream epiphenomena or secondary to a degenerative process.

Response: We really appreciate your constructive and important comments. To our knowledge, the expression changes of circRNAs have only been reported in association with the change of ALS pathogenesis-related molecules, but never reported as a factor directly associated with pathogenesis of ALS. Therefore, we think it is too early to judge whether the change in the expression level of circRNAs plays an etiological role in ALS pathogenesis. For clarity to the readers, we have modified the sentences as follows and added a reference (Qin C, et al. Front Neurosci, 2022).

Page 6-7, line 272-279 in the revised manuscript

“Taken together, although changes in expression of circRNAs have been proposed to be associated with the ALS pathogenesis-related molecular change, direct evidence suggesting the role of expression alteration of circRNAs in ALS pathogenesis remains elusive. Moreover, although rno_circ_013017 inhibits motor neuron apoptosis in rat with spinal cord injury [123], the role of circRNA in motor neuron biology in healthy elderly people or ALS patients has not been reported. Further knowledge on the etiological roles of circRNAs in motor neuron death in ALS will elucidate the interplay of several circRNAs in brain aging and neurodegeneration.”

On the other hand, as described in section 3.2 in the revised manuscript, dysregulation of RNA editing is not a downstream epiphenomen or secondary to a degenerative process but plays an etiological role in ALS pathogenesis. For readers’ better understanding, we have modified the sentences as follows and added Table 1 and a relevant reference (Suzuki H and Matsuoka M, J Neurochem 2021). Although you have kindly requested tables summarizing the roles of circRNA and RNA editing in ALS etiology, robust evidence suggesting that circRNAs have an etiological role in ALS pathogenesis has been scanty and inconclusive. Please allow us to present the evidence suggestive of the roles of circRNA in the text and summarize the etiological role of dysregulation of RNA editing in sporadic ALS and some forms of familial ALS in Table 1.

Page 7-8, line 324-327 in the revised manuscript

“Collectively, as RNA editing plays a pivotal role in brain aging and in the pathogenesis of sporadic ALS, and some forms of familial ALS, dysregulation of RNA editing might provide clues for the identification of biomarkers and serve as a potential therapeutic target for both brain aging and ALS.”

  1. Please keep the focus of the review on motor neuron biology as much as possible. Sometimes in the text (for example lines 145-149) it is not at all clear how this information relates to ALS pathogenesis.

Response: Thank you very much for this relevant observation. In line with the reviewer’s comment, we have reviewed the reports on dysregulation of RNA editing in the motor neurons in section 3.2 in revised manuscripts. Since conclusive reports on the changes in circRNAs expression levels in motor neurons are lacking, the effects of these circRNAs expression changes on motor neuron biology remains unknown. Accordingly, we have modified sentences as follows and added a reference (Qin C, et al. Front Neurosci, 2022)

Page 5, line 211-216 in the revised manuscript

“Additionally, expression levels of several circRNAs are altered in induced pluripotent stem cell (iPSC)-derived motor neurons of patients with ALS carrying the FUSP525L mutation compared with those carrying FUSWT [89], although the pathogenic significance and effects on motor neuron biology of the circRNA expression changes in ALS patients carrying the FUS mutation remain unknown.”

“Page 6-7, line 272-279 in the revised manuscripts”

As this comment is related to previous one, please refer the response to reviewer 1’s comment 33.

  1. The discussion of the anti-excitotoxic therapy testing in ALS is surprisingly not well developed or missing key information. The authors should discuss the trials with memantine, mexiletine, and retigabine.

Response: Thank you for this comment. We have added the discussion on the trials with memantine, mexiletine, and retigabine. However, roles of these drugs in dysregulation of RNA editing or regulation of circRNAs have not been reported. We have added the following sentences and some related references (Wang R, et al. Eur J Neurosci, 2005; de Carvalho, et al. Amyotroph Lateral Scler, 2010; Weiss, M.D, et al. Neurology, 2016; Weiss, M.D, et al. Muscle Nerve 2021; Shibuya K, et al. Amyotroph Lateral Scler Frontotempral Degener, 2015; Wainger B.J, et al. Cell Rep, 2014; Wainger, B.J, et al. JAMA Neurol, 2021)

Page 11, line 441-450 in the revised manuscript

“Other potential ALS drugs targeting neuronal hyperactivity or excitotoxicity have been reported; Memantine, a non-competitive antagonist of N-methyl-D-aspartic acid (NMDA) receptors, has been shown to delay disease progression and prolong survival of SOD1 mutant ALS model mice [181], but have no therapeutic benefits for ALS patients [182]. Mexiletine, a sodium channel blocker, has been shown to inhibit neuronal hyperexcitability and reduce the frequency of muscle clump [183,184], but is ineffective in inhibition disease progression [185]. Ezogabine, an activator of Kv7 potassium channels, has been shown to reduce neuronal excitability in vitro and in vivo [186,187]. There is no report, however, indicating the association between these candidate drugs and dysregulation of RNA editing or circRNAs.”

  1. Because of their work on this, the authors should better note that there were important issues with perampanel trial in that it was not well tolerated by the patients with potential detrimental effect.

Response: Thank you very much for appraising us about this important information. Although perampanel improved muscle strength in lower limbs, the high-dose perampanel participants had higher dropout rates than low-dose perampanel participants. The reason that the clinical trial failed to ameliorate disease progression includes the inhibitory effects on physiological function of AMPA receptors in other CNS neurons, and effects on molecules other than AMPA receptors, such as voltage-gated sodium channel, M-type potassium currents, and several kinases. The clinical trial of perampanel suggests that AMPA receptor antagonists that selectively inhibit Ca2+-permeable AMPA receptors but preserve physiological activity of AMPA receptors in general may become a candidate drug for ALS. For clarity to the readers, we have modified the sentences as follows and have added references (Lai M.C, et al. Biomolecules, 2019; Kim J.E, Front Cell Neurosci, 2019).

Page 10-11, line 430-434 in the revised manuscript

“Since non-AMPA antagonistic function, such as modulation of voltage-gated sodium channel and regulation of several kinases, with perampanel [179,180], and a dose-dependent increase in drop-out cases due to serious adverse effects may have influenced the results of the phase 2 trial, more selective AMPA receptor antagonists with lesser side effects are required.”

  1.  The text in general needs some careful editing because of awkward structure.   For example, lines 98-99, 117-122, 

Response: We really appreciate for pointing this out. After revising our manuscript to address the reviewers’ comments, we have had it rechecked by a native speaker of English. For clarity, we have modified the sentences as follow.

Page 3, line 103-109 in the revised manuscript

“Although there have been many comprehensive reviews that demonstrated circRNAs or RNA modification associated with age-related neurodegenerative diseases including ALS [48], no reviews have described the role of aging, circular RNAs, and RNA editing in pathogenesis of ALS. To the best of our knowledge, this is the first review which describes age-related alterations in circRNAs and RNA editing, focusing on their potential roles in neurodegeneration and the pathogenesis of ALS, and on the possibility of circRNAs and the dysregulation of RNA editing as biomarker candidates and therapeutic targets for ALS.”

Page 3, line 123-128 in the revised manuscript

“CircPVT1, a circular form of an exon of PVT1, influences cellular senescence and neurodegeneration by changing the expression levels of let-7 and miR-199-5a: let-7 regulates immune response, autophagy, and apoptosis [52-54], whereas  miR199-5a regulates expression levels of sirtuin1 (SIRT1) mRNA, which is associated with brain aging and neurodegeneration resulting from the dysregulation of mitochondrial energy metabolism [55-58].”

In addition to the above comments, all spelling and grammatical errors pointed out by the reviewers have been corrected.

We hope that you find our revised manuscript worthy of publication in the cells, and I thank you in advance for your consideration.

Sincerely,

Takashi Hosaka

Department of Neurology, Division of Clinical Medicine, Faculty of Medicine, University of Tsukuba, Tsukuba, Ibaraki, 305-8575, Japan

Tel. +81-29-853-3224

Reviewer 2 Report

1)      I have a problem with this manuscript. Although it is PDF, the figures covered their legends. I cannot see the first 2-3 lines of the legends in the whole of the figures.

2)       ALS is caused by many factors. Mutation in the C9orf72 gene and various epigenetic errors have roles in causing ALS.

https://www.ncbi.nlm.nih.gov/pmc/articles/PMC6331271/

I would recommend drawing a graphical abstract. APL should be in the middle and cause agents to be surrounded.

Adding an appropriate graphical abstract absorbs more audience.

3)    The authors should describe of biogenesis of circular RNAs. Moreover, section 5.1 is very short. There are many papers published in this area. Like

https://www.ncbi.nlm.nih.gov/pmc/articles/PMC7084402/

https://www.cell.com/molecular-therapy-family/methods/pdf/S2329-0501(20)30114-5.pdf

https://molecular-cancer.biomedcentral.com/articles/10.1186/s12943-017-0663-2

4)    I really would like to know what is the novelty of this review since there are many comprehensive reviews have been published.

5)    Histone Modifications and Chromatin Remodeling Enzymes in ALS have not been discussed in the review either in text or figure.

6)    Aging is not an etiology of ALS since there are plenty of old people does not have any neurodegeneration diseases.  During aging, the risk of cellular dysfunctionality becomes higher. If the cell’s function is knockdown specifically, these diseases appear.   

7)    I strongly recommend adding a table at the end of section 5.2 to summarize the therapeutic agent. In this table, the target of each agent, whether is FDA approved or not, respected reference must be mentioned. 

Author Response

Point-by-Point Responses to Reviewers’ comments

○Reviewer 2

1)  I have a problem with this manuscript. Although it is PDF, the figures covered their legends. I cannot see the first 2-3 lines of the legends in the whole of the figures. 

Response: We really apologized to the unexpected problem.

2) ALS is caused by many factors. Mutation in the C9orf72 gene and various epigenetic errors have roles in causing ALS.

https://www.ncbi.nlm.nih.gov/pmc/articles/PMC6331271/ 

I would recommend drawing a graphical abstract. APL should be in the middle and cause agents to be surrounded. 

Adding an appropriate graphical abstract absorbs more audience. 

Response: Thank you and we really appreciate the reviewer’s valuable comments. To facilitate reader’s understandings and interests, we have prepared a graphical abstract.

3) The authors should describe of biogenesis of circular RNAs. Moreover, section 5.1 is very short. There are many papers published in this area. Like

https://www.ncbi.nlm.nih.gov/pmc/articles/PMC7084402/ 

https://www.cell.com/molecular-therapy-family/methods/pdf/S2329-0501(20)30114-5.pdf 

https://molecular-cancer.biomedcentral.com/articles/10.1186/s12943-017-0663-2 

Response: Thank you. We appreciate your valuable comments. For readers’ better understanding, we have added the following sentences in INTRODUCTION and some references in order to facilitate clarity and understanding (Meng S, et al. Mol Cancer, 2017; Ren S, et al. Mol Ther Methods Clin Dev, 2020). Also, to facilitate reader’s understanding of section 4.1 (Potential biomarker candidates for ALS) in revised manuscripts, we have modified the sentences as follows and added some references (Campos-Melo D, et al. Mol Brain, 2013; Vrabec K, et al. Front Mol Neurosci, 2018; Ravnik-Glavač M, et al. Int J Mol Sci, 2020).

Page 2, line 60-66 in the revised manuscript

“Circular RNAs (circRNAs) are single-stranded RNA molecules circularized by the covalent joining of the 3′-end to the 5′-end, known as back-splicing [15,16], and divided into three classes; circular intronic RNA (ciRNA), exonic circRNA (ecircRNA), and exon-intron circRNA (EIciRNA) [17]. CiRNAs are produced by canonical splicing and escape from debranching enzyme, whereas ecircRNA and EIciRNAs are generated by back-splicing with the assistance of complementary sequences between flanking intron and RNA-binding proteins (RBPs) [18].”

Page 9, line 376-380 in the revised manuscript

“Among them, hsa_circ_0000567 and hsa_circ_0063411 contain binding sites for miR-9 and miR-641, respectively, the expression levels of which are shown to change in ALS patients [157,158], and hsa_circ_0023919, hsa_circ_0063411, and hsa_circ_0088036 are potential diagnostic biomarkers because of their high sensitivity and specificity to ALS [156,159].”

4) I really would like to know what is the novelty of this review since there are many comprehensive reviews have been published. 

Response: You have raised an important point here. However, we believe that as no previous reviews have focused on the effects of aging on dysfunction of RNA editing and circRNA, the novelty of this review is the focus on the association between age-related dysregulation of RNA editing and expression change of circRNAs in relation to pathogenesis of ALS. For readers’ better understanding, we have added the following sentences in INTRODUCTION and a reference (Jiapaer Z, et al. Aging cell, 2022).

Page 3, line 103-109 in the revised manuscript

“Although there have been many comprehensive reviews that demonstrated circRNAs or RNA modification associated with age-related neurodegenerative diseases including ALS [48], no reviews have described the role of aging, circular RNAs, and RNA editing in pathogenesis of ALS. To the best of our knowledge, this is the first review which describes age-related alterations in circRNAs and RNA editing, focusing on their potential roles in neurodegeneration and the pathogenesis of ALS, and on the possibility of circRNAs and the dysregulation of RNA editing as biomarker candidates and therapeutic targets for ALS.”

5) Histone Modifications and Chromatin Remodeling Enzymes in ALS have not been discussed in the review either in text or figure. 

Response: As the reviewer correctly pointed out, increasing numbers of reports describe the role of alteration of gene expression due to epigenetic modification in ALS pathogenesis. It is reported that HDAC inhibitors increase, but not decrease, the expression of ADAR2, their significance in ALS pathogenesis, in which dysregulation of RNA editing caused by ADAR2 downregulation plays a pivotal role, remains elusive. Therefore, to facilitate readers’ understanding, we have added sentences as follows and some references (Han Q, et al. Biochem J, 2008; Barznegar M, et al. Cell Biochem Funct, 2022; Valle C, et al. Cell Death Dis, 2014; Chen K, et al. ACS Chem Neurosci, 2018; Xie B, et al. Gene, 2019; Wang W, et al. Cell Death Dis, 2021; Jun M.H, et al. Sci Rep, 2017; Ma J, et al. J Bioenerg Biomembr, 2021; Bennett S.A, et al. Transl Res, 2019; Uchida H, et al. Neuroreport, 2015), but not added in the figures.

Page 5-6, line 230-240 in the revised manuscript

“Moreover, some circRNAs influence the expression levels of epigenetic enzymes; circELP3 (hsa_circ_0001785) regulates the expression of elongator acetyltransferase complex subunit 3 (ELP3) [98], a histone acetyltransferase, that has an association with motor neuron degeneration via regulating heat shock protein [99]. CircTHBS2 (hsa_circ_0078710) and circSIRT1 (hsa_circ_0093844) regulate the expression levels of histone deacetylases (HDACs) and SIRT1, respectively [100,101], both of which have an association with epigenetic modification in ALS pathogenesis [102,103]. CircLRP6 regulates the expression level of protein arginine N-methyltransferase (PRMT1), a histone methyltransferase [104], that rescues neurite growth in FUSR521Cmice [105]. However, there is no evidence indicating the direct roles of these circRNA-related expression changes in neurodegeneration or ALS pathogenesis.”

Page 7, line 311-317 in the revised manuscript

“Recently, sodium phenylbutyrate/ursodoxicoltaurine, a combination drug of HDAC inhibitor and activator of mitochondria bioenergetic, was approved for an ALS treatment by Health Canada and FDA in USA [10]. This suggests that epigenetic modification such as histone modification and chromatin remodeling enzyme may contribute to ALS pathogenesis [142]. Intriguingly, inhibition of HDACs, by treatment with trichostatin A increases expression level of ADAR2 mRNA [143].”

6) Aging is not an etiology of ALS since there are plenty of old people does not have any neurodegeneration diseases.  During aging, the risk of cellular dysfunctionality becomes higher. If the cell’s function is knockdown specifically, these diseases appear.   

Response: We would like to thank the reviewer for pointing out this important aspect. As pointed out by reviewer, aging is not a direct pathogenetic factor of ALS. However, age-related dysregulation of RNA editing may accelerate age-independent ADAR2 downregulation mechanism in ALS motor neurons and increase circRNAs expression. To avoid the reader’s misunderstanding and facilitate the readers’ understanding, we have produced a graphic abstract and modified a following sentence and Figure 3.

Page 8, line 332-334 in the revised manuscript

“Aging is a major risk factor for neurodegenerative diseases, and dysregulation of circRNAs and RNA editing with aging are associated with neurodegenerative diseases including ALS.”

7) I strongly recommend adding a table at the end of section 5.2 to summarize the therapeutic agent. In this table, the target of each agent, whether is FDA approved or not, respected reference must be mentioned. 

Response: We really appreciate the reviewer’s important and valuable comments. However, there are only a few established therapeutic agents based on dysregulation of RNA editing and no established agents on circRNAs have been reported. Moreover, there are no FDA approved agents of treatments for ALS targeting circRNA, RNA editing or aging. Therefore, we think these agents may not be worth listing in a table.

In addition to the above comments, all spelling and grammatical errors pointed out by the reviewers have been corrected.

We hope that you find our revised manuscript worthy of publication in the cells, and I thank you in advance for your consideration.

Sincerely,

Takashi Hosaka

Department of Neurology, Division of Clinical Medicine, Faculty of Medicine, University of Tsukuba, Tsukuba, Ibaraki, 305-8575, Japan

Tel. +81-29-853-3224

Round 2

Reviewer 1 Report

The revisions are appropriate and acceptable.  Thnak you.

Reviewer 2 Report

The authors answered my question carefully. I recommend accepting.